# Anatomy of Garnet from the Nanminghe Skarn Iron Deposit, China: Implications for Ore Genesis

**Chen-Tao Ruan** [1], **Xiao-Yan Yu** [1,*], **Shang-Guo Su** [2], **M. Santosh** [2,3] and **Li-Jie Qin** [1]

[1] School of Gemology, China University of Geosciences Beijing, 29 Xueyuan Road, Beijing 100083, China; rct2900580539@163.com (C.-T.R.); 2009200043@cugb.edu.cn (L.-J.Q.)

[2] School of Earth Sciences and Resources, China University of Geosciences Beijing, Beijing 100083, China; susg@cugb.edu.cn (S.-G.S.); santosh@cugb.edu.cn (M.S.)

[3] Department of Earth Science, University of Adelaide, Adelaide, SA 5005, Australia

* Correspondence: yuxy@cugb.edu.cn

**Abstract:** Garnet is a common constituent of skarn type iron deposits and can be used to derive potential information on the genesis of skarn type deposits. Here, we investigate the petrologic, spectroscopic, and geochemical characteristics of garnet from the Nanminghe skarn iron deposit in China to elucidate the formation process, growth environment, and genesis. We employ a combination of multiple techniques including petrography, Infrared spectroscopy (IR), X-ray powder diffraction (XRD), Raman spectrum, electron microprobe, and LA-ICP-MS. The primary mineral assemblage in the skarn is garnet–diopside–magnetite–quartz–calcite–pyrite. The garnet occurs as granular aggregates or veins, and generally shows a combination form bounded by dodecahedral faces {110} and trapezohedron faces {211}. Oscillatory zoning and abnormal extinction of garnet are also noted. We identify at least three stages of garnet growth, with a gradual decrease in the iron content from early to late stage, accompanied by the precipitation of magnetite. Regarding the rare earth distribution model, the Nanminghe garnet is generally in the right-dipping mode enriched in LREE and depleted in HREE, which may be mainly controlled by adsorption. Major and trace elements of different generations of garnet suggest that the garnet in the iron skarn crystallized under high oxygen fugacity and is of hydrothermal origin.

**Keywords:** andradite; iron skarn; hydrothermal origin; oscillatory zonation





## 1. Introduction

Garnet is a widely distributed mineral with the general chemical formula $^{[8]}X_3^{[6]}Y_2^{[4]}Z_3^{[4]}O_{12}$, where X, the eight-coordinated dodecahedral site, is usually $Ca^{2+}$, $Mn^{2+}$, $Fe^{2+}$, and/or $Mg^{2+}$; Y, the six-coordinated octahedral site, is generally $Al^{3+}$, $Fe^{3+}$, $Cr^{3+}$, and/or $V^{3+}$; and Z, the four-coordinated tetrahedral site, is commonly $Si^{4+}$ [1,2]. Isomorphic substitution is prevalent in garnet, and ion substitution may occur at X and Y positions, or even in the Z position. Garnets are usually divided into two series: aluminum series (Y position is $Al^{3+}$) and calcium series (X position is $Ca^{2+}$). The general end-member composition of the two series are as follows: aluminum series—pyrope, almandine, and spessartine; calcium series—grossular, andradite, and uvarovite, and most garnets in nature are solid solutions of two or more end-member composition [1,3]. Garnet is common in magmatic, hydrothermal, and metamorphic rocks [4,5]. Gem quality garnet deposits are widely distributed in China [6]. Skarn deposits represent a globally significant source of Cu, Fe, Pb, Zn, W, Ag, and Au, and skarn garnets could be formed by both contact metamorphism and hydrothermal alteration of carbonate-bearing rocks [7,8]. Garnet is one of the main rock-forming minerals of skarns [9,10], and hosts important genetic information of the hydrothermal fluid evolution and ore-forming environment [11,12].

The Nanminghe iron deposit in the North China Craton is an important skarn deposit, and its genesis remains controversial, with diverse models of contact metasomatic

origin [13,14] and magmatic origin [15]. Since garnet is the main mineral in this skarn deposit, we present results from a detailed study of the geochemistry and spectroscopic characteristics of garnet from this deposit to understand the physicochemical conditions and oxygen fugacity changes during the growth of garnet. Our results provide insights into garnet formation in skarn deposits with complex evolutionary history and can be used for understanding the genesis of these deposits and their exploration.

## 2. Geological Setting

The Handan–Xingtai district (Figure 1a), located in the southern Taihang Mountains in the central North China Craton (NCC) within the Trans-North China Orogen (TNCO) (Figure 1b), is one of the most important skarn-type iron ore fields in China, with proven reserves of 900 to 1000 Mt at an average of 40 to 55 wt.% Fe [14,16,17]. Wuan is located in the middle of the Handan–Xingtai district, to the east of the Taihang Mountain fault zone [14,16,18].

The major faults and folds in this area are showing a NE- and NNE-trend [14,15,18,19] (Figure 1b). The skarn iron deposits in the Wuan region occur along the contact zone between the Yanshanian diorite–monzonite and sedimentary carbonates of the Middle Ordovician, with magnetite as the primary ore mineral [13,16,17,19,20]. Iron skarn deposits of the Handan–Xingtai district formed during the early Cretaceous as two different pulses, spanning from 137.1 ± 1.5 to 134.1 ± 1.2 Ma and from 130.5 ± 0.9 to 128.5 ± 0.9 Ma, and the Nanminghe deposit is associated with the second mineralizing episode [17].

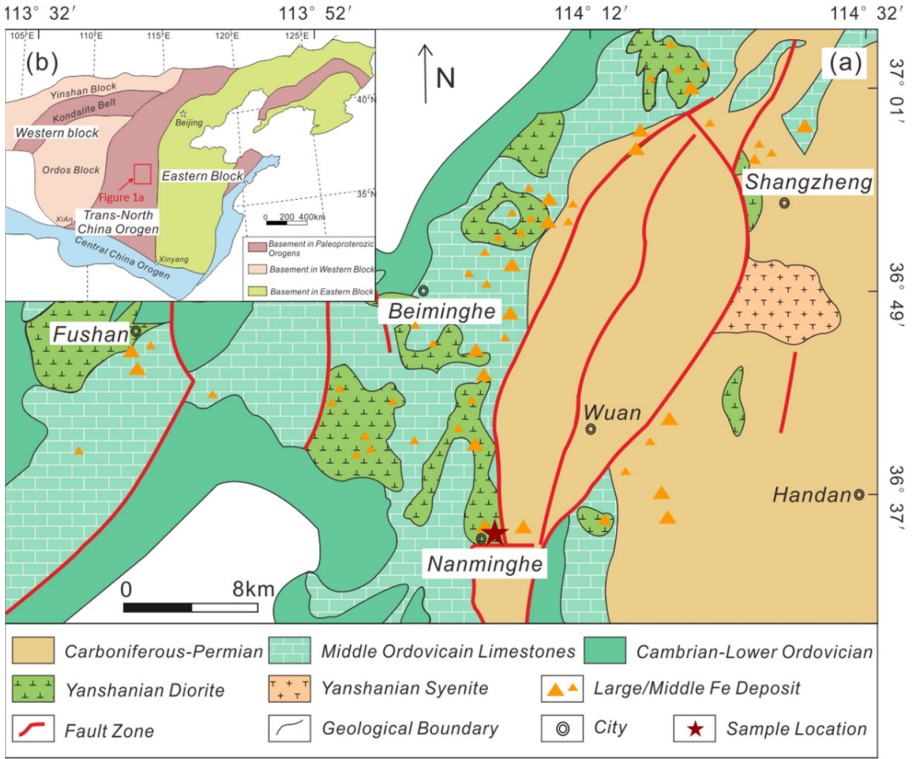

**Figure 1.** (**a**) Geologic map of Wuan district (modified from [14]); (**b**) Major tectonic units of the North China Craton (modified from [21]).

The Nanminghe iron deposit is located in Cishan Town, Wuan City, Hebei Province and is a concealed ore body with a buried depth of 80.6–837 m and an average thickness of 12.6 m [22]. It is located in the lower part of the coal mining area of the southern wing of Wannianjing, with a total of six ore bodies, of which the $Fe_6$ ore body is the main collection area [22]. Although the scale does not reach the standard of large iron ore, the Nanminghe iron ore is one of the important iron deposits in the Wuan region due to the

high grade of magnetite. The existing drilling data indicate that the top of the Nanminghe iron deposit is composed of Middle Ordovician limestones and the bottom is Yanshanian diorite, with the iron skarn occurring in the contact zone between these two. A previous zircon U-Pb geochronological study suggested that the dioritic intrusions were emplaced at $(136 \pm 2)$–$(129 \pm 1)$ Ma [17].

## 3. Materials and Methods

### 3.1. Materials

Within the garnet skarn belt of the $Fe_6$ ore body, at a depth of about 300 m underground, 24 representative samples collected were partially ground into powder with a grain size under 200 mesh. Seven representative samples were cut into polished thin cross-sections in the laboratory of China University of Geosciences (Beijing). A detailed petrographic study was carried out using a Nanjing Baoguang GI-MP22 gem microscope and Olympus DP74 petrographic microscope at the China University of Geoscience Beijing (CUGB).

### 3.2. Analytical Methods

#### 3.2.1. Infrared Spectroscopy

Infrared spectroscopy was performed at the gem appraisal center of the Peking University using a Bruker Tensor 27 Fourier-transform infrared spectrometer. The experimental conditions were set as follows: electric power 210–230 W, power frequency 50–60 Hz, resolution 4 $cm^{-1}$, scan time 16 s. Pellets of dry KBr, weighing 150 mg with a sample/matrix weight ratio of 1:150, were analyzed for the infrared spectra from 4000 to 400 $cm^{-1}$ in the ambient conditions.

#### 3.2.2. Raman Spectroscopy

Raman spectroscopy was performed at China University of Geoscience Beijing (CUGB) using aHORIBA LabRAM HR-Evolution Raman spectromete. The Raman spectrum was recorded in the 100–2000 $cm^{-1}$ range with one scan and a 4 $cm^{-1}$ resolution, with an excitation light source wavelength of 532 nm.

#### 3.2.3. X-ray Powder Diffraction (XRD)

X-ray powder diffraction was performed at the Beijing Center for Physical and Chemical Analysis using a Bruker D8 advance X-ray diffractometer. The experimental conditions were as follows: X-ray source—Cu target, voltage—40 kV, current—40 mA, using a step scan, step—0.02°/step, scan speed—0.1 s/step, and scan range of 5–90°.

#### 3.2.4. Electron Microprobe

The elemental concentrations and distributions within microstructural domains were measured by a JEOL JXA-8100 electron microprobe at the Electron Probe Laboratory of Chinese Academy of Geological Sciences. The experimental conditions were as follows: accelerating voltage—15 kV, beam current—20 nA, beam diameter—5 μm. A total of 11 elements—Na, Mg, Al, Si, Cr, Mn, Fe, Ni, K, Ca, and Ti, were tested using Na (albite), Al (corundum), Si (diopside), Mn (synthetic MnO), Fe (hematite), Ca (diopside), Cr (synthetic $Cr_2O_3$), Ni (synthetic NiO), Ti (rutile), K (orthoclase), and Mg (diopside) as standards. All microprobe data were corrected by the ZAF program and the end member components were calculated by the Excel program Geokit.

#### 3.2.5. LA-ICP-MS

The analysis of trace elements was performed at Wuhan SampleSolution Co., LTD using the Agilent 7900 ICP-MS coupled with a GeoLas HD 193 nmPro laser ablation system. During laser denudation, He was used as the carrier gas and Ar as the compensation gas to adjust the sensitivity. The laser spot diameter was 44 μm, the frequency was 5 Hz, and the laser energy was 80 mJ. International standard reference materials BHVO-2G, BCR-2G, and BIR-1G were used to calibrate trace element contents. Data processing was performed using

ICPMSDATACAL10.8 software. Details of the specific analysis conditions and procedures follow the description of Liu et al. (2008) [23].

## 4. Results

### 4.1. Garnet Morphology and Inclusions

The garnet in the skarn ranges in size from 0.1 to 20 mm and occurs as granular aggregates or veins. They are brown–dark brown and mostly euhedral–subhedral. More than 70% of the garnet grains show the combination form bounded by dodecahedral faces {110} and trapezohedron faces {211} (Figure 2a), and the dodecahedral simplex is also visible (Figure 2b). Combination striations and cracks are visible on the crystal face. Symbiotic/associated minerals such as hematite, pyrite, diopside, and calcite can be seen in the garnet samples. Pore structure is common in hand specimens (Figure 2c), and minerals such as garnet, calcite, and quartz can be seen around the pores. Crystal patterns can be seen on the rhombic dodecahedron face (Figure 2d), and the rhombic growth pits and mounts parallel the crystal edges. The solid inclusion shown in Figure 2e is also garnet, which is relatively frequent for this mineral. Quartz, diopside, magnetite, and other solid inclusions can also be seen. The gas–liquid fluid inclusions (Figure 2f) are very common in the garnet samples and are too small to be measured by instruments for the composition of gases and liquids.

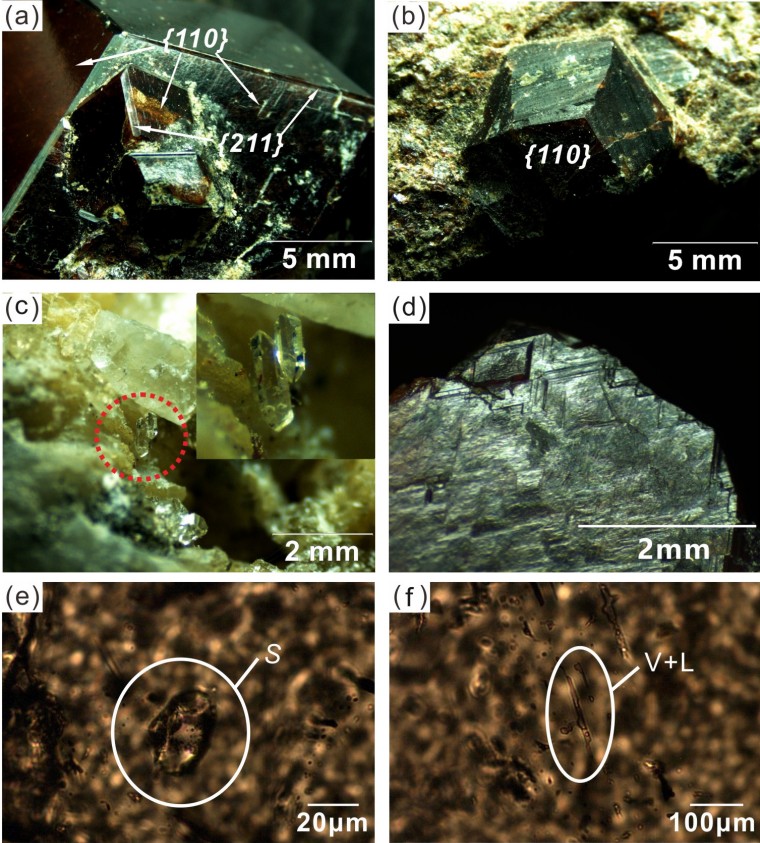

**Figure 2.** Representative photomicrographs of garnet samples. (**a**) The combination form of garnet bounded by dodecahedral faces {110} and trapezohedron faces {211}; (**b**) Dodecahedral simplex of garnet; (**c**) Quartz crystal around the pore (dashed line); (**d**) Crystal patterns on dodecahedral face under gem microscope; (**e**,**f**) The solid inclusion and gas–liquid inclusion (solid line) under Raman microscope.

### 4.2. Petrography

Under plane polarized light, the garnet grains show a euhedral–subhedral granular structure with positive high relief and randomly distributed microcrack in a light yellow–

light brown color. The microscopic rhombic growth pits on the rhombic dodecahedron face are visible under the microscope (Figure 3a). According to petrographic features, at least three stages of garnet growth can be identified. (1) Stage I: Isotropic and subhedral garnets occurring as granular aggregates with later formed magnetite filling the cracks or the intergranular pace, metasomatizing the garnet formed in the earlier stage (Figure 3b). (2) Stage II: Euhedral–subhedral garnet with relatively complete crystal form, mainly consisting of the combination form bounded by dodecahedral faces {110} and trapezohedron faces {211}, with coeval diopside (Figure 3c). The subhedral garnets forming a granular aggregate are isotropic with uniform extinction, whereas the euhedral garnets usually have isotropic cores with abnormal extinction (Figure 3d left) and anisotropic rims with oscillatory zoning (Figure 3d, right), with irregular boundary (dashed line) (Figure 3d, right) suggesting external stress regime during growth [24]. The epitaxial growth of high-index metastable {211} faces on stable {110} faces may reflect the change in the growth rate of garnet [24,25] (Figure 3e left). Some crystal growth features are similar to those described by [24], such as the deformation lamellae perpendicular to the crystal face (Figure 3e, right) and the irregular boundary between sectors (Figure 3e, right), which is the result of externally induced local strain [25]. (3) Stage III: The isotropic and vein garnets (dashed line) (Figure 3f), cut through early light yellow euhedral garnets (stage II), are light brownish-yellow and visible on the thin section.

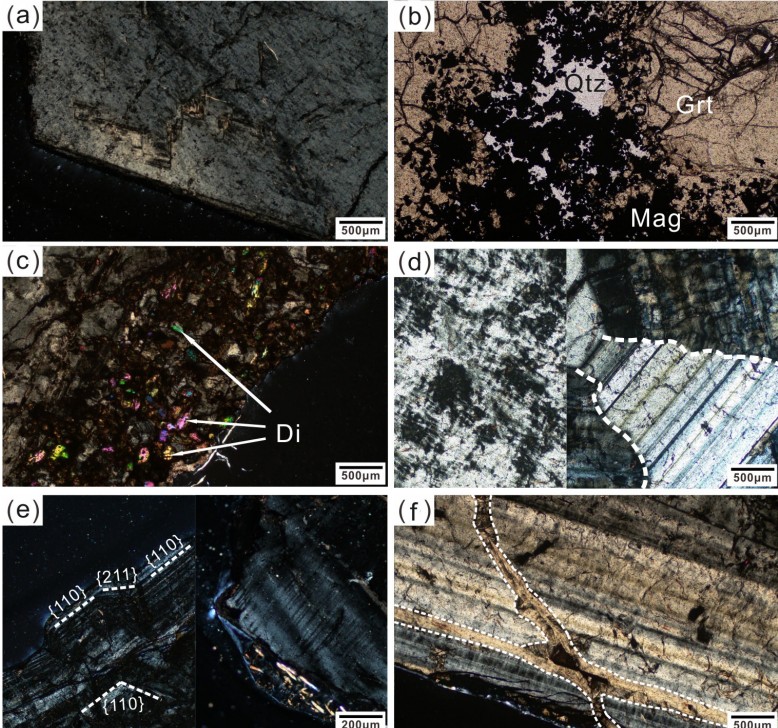

**Figure 3.** Representative photomicrographs under crossed polarized light XPL and under plane polarized light PPL. (**a**) The microscopic rhombic growth pits on the rhombic dodecahedron face (XPL); (**b**) Isotropic and subhedral stage I garnet and associated minerals, magnetite and quartz (PPL); (**c**) Anisotropic stage II garnet with granular diopside (XPL); (**d**) Abnormal extinction, such as patch extinction (left), sector extinction, and oscillatory zonation (right) (XPL). Note the irregular boundary (dashed line); (**e**) The {211} faces appear on the outer edge of the {110} faces (left) and the lamellae perpendicular to crystal faces (right) (XPL); (**f**) Stage III vein garnet (dashed line) cut through stage II garnet (XPL). Mineral abbreviations: Di—diopside; Mag—magnetite; Grt—garnet; Qtz—quartz.

*4.3. Infrared Spectra and X-ray Powder Diffraction*

The absorption bands of all the examined samples are characteristic infrared absorption spectra of garnet, located between grossular and andradite, which are closer to andradite.

In the previous studies, the absorption bands due to structural $OH^-$ of garnets are mainly located at about 3550–3560 $cm^{-1}$ and 3610–3680 $cm^{-1}$ [26–28], whereas this absorption peak is not observed in garnet from the Nanminghe deposit in this study. We thus infer that the Nanminghe garnets do not contain any structural water, or the content is extremely low and difficult to detect. The experimental data, analyzed using Jade software, show that the diffraction peaks of the garnet samples are consistent with the standard diffraction peaks of andradite (Figure 4a), indicating that the main component of the garnet is andradite. The associated minerals include magnetite, quartz, calcite, and diopside (Figure 4b–d).

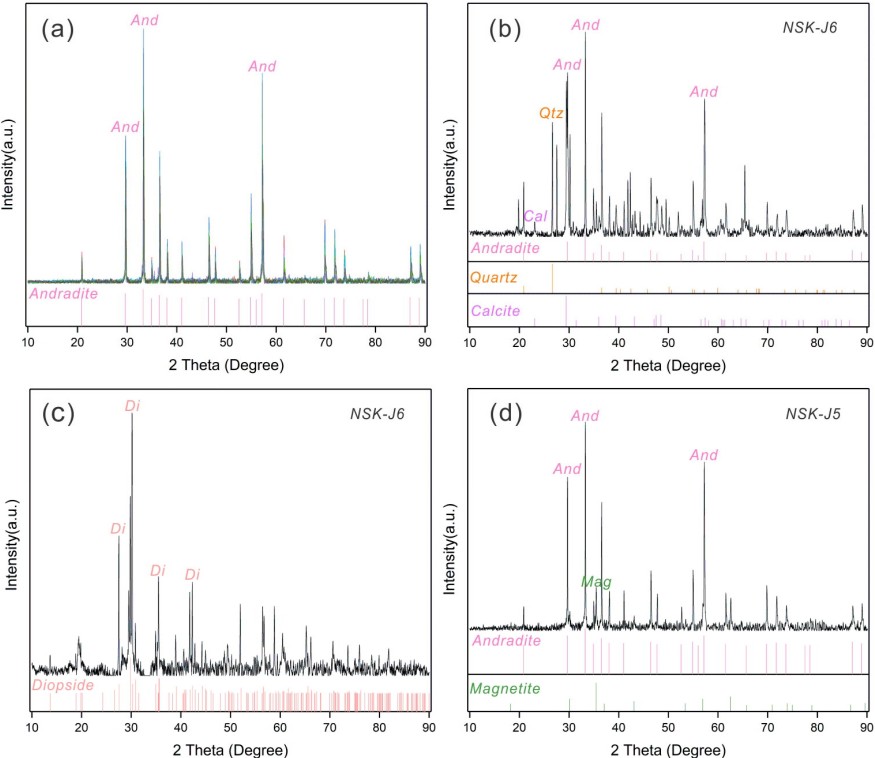

**Figure 4.** Representative XRD of garnet and wall rock samples. (**a**) Andradite; (**b**) andradite + quartz + calcite; (**c**) diopside; (**d**) andradite + magnetite.

### 4.4. Major Element Geochemistry

The major element results and the calculated garnet end-member compositions are shown in Table 1. The content of $SiO_2$, $Al_2O_3$, $FeO$, and $CaO$ of garnets in different stages shows minor differences. We combined the three stages of data and found that andradite end-member composition in stage I ranges from 96.43% to 98% (average 97%), from 65% to 95% in stage II (average 83%), and from 86% to 90% in stage III (average 88%). The grossular end-member composition in stage I ranges from 0 to 8%, from 2% to 33% in stage II, and from 6% to 10% in stage III. Twenty-two points from the core to the edge of sample NSK-J12 were selected for analysis. Figure 5a shows the backscattered electron image of oscillating bands in garnet, and Figure 5b shows the changes in the main elements of Al, Fe, Mn, and Ca. The contents of Al and Mn in the dark band are obviously higher than in the light grey band, while the opposite occurs for Fe. The grossular end-member composition is higher in the dark band; with the transition from dark gray to light gray, the grossular end-member decreases and andradite end-member composition increases accordingly (Figure 5c). The major elements and end-member compositions of the core, except for Ca, are essentially unchanged (Figure 5b-c, point 18–22). The above features indicate that garnet grew in a stable environment in the early stage of growth, but conditions were later oscillating.

Table 1. Representative electron microprobe analyses for studied garnet with oscillatory zonation(wt.%).

| Sample | NSK-J5-2-1 | NSK-J5-2-3 | NSK-J5-q3 | NSK-J5-q4 | NSK-J5-3-1 | NSK-J5-1 | NSK-J2-1-1 | NSK-J2-1-2 | NSK-J23-1-7 | NSK-J23-1-8 | NSK-J23-1-9 |
|---|---|---|---|---|---|---|---|---|---|---|---|
| Description | Isotropic | Isotropic | Isotropic | Isotropic | Isotropic | Isotropic | Isotropic | Isotropic | Anisotropic | Anisotropic | Anisotropic |
| Stage | I | I | I | I | I | I | II | II | II | II | II |
| $SiO_2$ | 36.37 | 36.13 | 35.90 | 36.21 | 36.18 | 35.66 | 36.45 | 34.85 | 37.49 | 37.35 | 37.69 |
| $TiO_2$ | 0.03 | 0.00 | 0.03 | 0.02 | 0.00 | 0.00 | 0.01 | n.d. | n.d. | 0.01 | n.d. |
| $Al_2O_3$ | 0.71 | 0.57 | 0.49 | 0.56 | 0.38 | 0.41 | 1.22 | 0.80 | 5.65 | 3.40 | 7.25 |
| $Cr_2O_3$ | 0.01 | 0.04 | 0.00 | 0.00 | 0.00 | 0.02 | n.d. | n.d. | n.d. | 0.01 | n.d. |
| FeO* | 27.74 | 27.90 | 28.35 | 28.33 | 28.21 | 28.13 | 27.00 | 27.20 | 21.06 | 23.93 | 19.34 |
| MnO | 0.78 | 0.77 | 0.65 | 0.73 | 0.66 | 0.69 | 0.85 | 0.57 | 0.81 | 0.67 | 0.88 |
| MgO | 0.16 | 0.17 | 0.05 | 0.04 | 0.06 | 0.03 | 0.21 | 0.04 | 0.02 | 0.08 | 0.01 |
| CaO | 32.14 | 32.35 | 32.84 | 32.56 | 32.71 | 32.35 | 32.54 | 32.93 | 33.59 | 33.96 | 34.22 |
| Total | 97.94 | 97.94 | 98.31 | 98.46 | 98.19 | 97.29 | 98.28 | 96.40 | 98.62 | 99.41 | 99.38 |
| Si | 3.030 | 3.015 | 2.992 | 3.010 | 3.014 | 3.003 | 3.020 | 2.964 | 3.038 | 3.032 | 3.018 |
| Ti | 0.002 | 0.000 | 0.002 | 0.001 | 0.000 | 0.000 | 0.001 | 0.000 | 0.000 | 0.001 | 0.000 |
| Al | 0.070 | 0.056 | 0.048 | 0.055 | 0.037 | 0.041 | 0.119 | 0.081 | 0.540 | 0.325 | 0.684 |
| Cr | 0.001 | 0.003 | 0.000 | 0.000 | 0.000 | 0.001 | 0.000 | 0.000 | 0.000 | 0.000 | 0.000 |
| $Fe^{3+}$ | 1.908 | 1.932 | 1.955 | 1.938 | 1.953 | 1.956 | 1.867 | 1.935 | 1.427 | 1.624 | 1.295 |
| $Fe^{2+}$ | 0.024 | 0.016 | 0.021 | 0.032 | 0.013 | 0.025 | 0.004 | 0.000 | 0.000 | 0.000 | 0.000 |
| Mn | 0.055 | 0.055 | 0.046 | 0.051 | 0.046 | 0.049 | 0.060 | 0.041 | 0.055 | 0.046 | 0.059 |
| Mg | 0.020 | 0.022 | 0.007 | 0.005 | 0.007 | 0.003 | 0.026 | 0.005 | 0.002 | 0.009 | 0.001 |
| Ca | 2.869 | 2.893 | 2.933 | 2.900 | 2.920 | 2.919 | 2.889 | 3.002 | 2.916 | 2.954 | 2.936 |
| Uvarovite | 0.03 | 0.13 | 0.00 | 0.00 | 0.00 | 0.05 | 0.00 | 0.01 | 0.00 | 0.02 | 0.00 |
| Andradite | 96.43 | 96.80 | 97.57 | 97.04 | 97.79 | 97.36 | 94.01 | 95.22 | 71.98 | 80.97 | 64.82 |
| Pyrope | 0.68 | 0.73 | 0.22 | 0.17 | 0.23 | 0.11 | 0.87 | 0.18 | 0.08 | 0.30 | 0.04 |
| Spessartine | 1.86 | 1.83 | 1.52 | 1.71 | 1.55 | 1.65 | 2.00 | 1.34 | 1.86 | 1.54 | 1.98 |
| Grossular | 0.18 | 0.00 | 0.00 | 0.00 | 0.00 | 0.00 | 2.98 | 3.25 | 26.08 | 17.17 | 33.16 |
| Almandine | 0.82 | 0.52 | 0.69 | 1.08 | 0.43 | 0.83 | 0.14 | 0.00 | 0.00 | 0.00 | 0.00 |

| Sample | NSK-J22-3-1 | NSK-J22-1-9 | NSK-J22-1-2 | NSK-J12-1-3 | NSK-J24-2-1 | NSK-J24-2-4 | NSK-J24-2-5 | NSK-J22-l1 | NSK-J22-l3 | NSK-J22-l5 |
|---|---|---|---|---|---|---|---|---|---|---|
| Description | Anisotropic | Anisotropic | Anisotropic | Anisotropic | Anisotropic | Anisotropic | Anisotropic | Anisotropic | Anisotropic | Anisotropic |
| Stage | II | II | II | II | II | II | III | III | III | III |
| $SiO_2$ | 35.92 | 37.34 | 36.57 | 36.56 | 36.58 | 37.09 | 37.01 | 36.88 | 36.89 | 36.69 |
| $TiO_2$ | 0.02 | 0.02 | 0.04 | 0.03 | n.d. | 0.02 | 0.12 | 0.24 | 0.12 | 0.14 |
| $Al_2O_3$ | 1.03 | 3.96 | 1.48 | 2.10 | 4.15 | 6.71 | 2.11 | 2.50 | 2.14 | 2.87 |

**Table 1.** *Cont.*

| Sample | NSK-J22-3-1 | NSK-J22-1-9 | NSK-J22-1-2 | NSK-J12-1-3 | NSK-J24-2-1 | NSK-J24-2-4 | NSK-J24-2-5 | NSK-J22-l1 | NSK-J22-l3 | NSK-J22-l5 |
|---|---|---|---|---|---|---|---|---|---|---|
| Description | Anisotropic | Anisotropic | Anisotropic | Anisotropic | Anisotropic | Anisotropic | Anisotropic | Anisotropic | Anisotropic | Anisotropic |
| Stage | II | II | II | II | II | II | III | III | III | III |
| $Cr_2O_3$ | 0.05 | 0.01 | n.d. | n.d. | 0.03 | 0.02 | 0.02 | n.d. | 0.03 | 0.01 |
| FeO* | 27.61 | 23.37 | 26.83 | 25.78 | 23.28 | 19.78 | 26.51 | 25.12 | 26.07 | 25.34 |
| MnO | 0.70 | 0.80 | 0.66 | 0.80 | 0.73 | 0.91 | 0.99 | 0.84 | 0.91 | 1.12 |
| MgO | 0.04 | 0.02 | 0.07 | 0.05 | 0.02 | 0.01 | 0.02 | 0.01 | 0.02 | 0.04 |
| CaO | 32.61 | 33.56 | 33.13 | 32.34 | 33.50 | 33.76 | 32.21 | 32.84 | 32.77 | 32.21 |
| Total | 97.98 | 99.08 | 98.80 | 97.65 | 98.30 | 98.29 | 98.98 | 98.42 | 98.94 | 98.40 |
| Si | 2.998 | 3.034 | 3.014 | 3.037 | 3.001 | 3.010 | 3.037 | 3.032 | 3.026 | 3.022 |
| Ti | 0.001 | 0.001 | 0.003 | 0.002 | 0.000 | 0.001 | 0.007 | 0.015 | 0.008 | 0.009 |
| Al | 0.101 | 0.379 | 0.144 | 0.205 | 0.401 | 0.641 | 0.204 | 0.242 | 0.207 | 0.278 |
| Cr | 0.003 | 0.001 | 0.000 | 0.000 | 0.002 | 0.001 | 0.001 | 0.000 | 0.002 | 0.000 |
| $Fe^{3+}$ | 1.896 | 1.588 | 1.845 | 1.769 | 1.596 | 1.343 | 1.766 | 1.727 | 1.769 | 1.701 |
| $Fe^{2+}$ | 0.031 | 0.000 | 0.004 | 0.022 | 0.001 | 0.000 | 0.054 | 0.001 | 0.020 | 0.044 |
| Mn | 0.049 | 0.055 | 0.046 | 0.056 | 0.051 | 0.062 | 0.069 | 0.059 | 0.063 | 0.078 |
| Mg | 0.005 | 0.003 | 0.009 | 0.006 | 0.003 | 0.001 | 0.002 | 0.002 | 0.002 | 0.005 |
| Ca | 2.916 | 2.921 | 2.925 | 2.878 | 2.944 | 2.936 | 2.832 | 2.893 | 2.880 | 2.843 |
| Uvarovite | 0.16 | 0.03 | 0.00 | 0.00 | 0.11 | 0.06 | 0.07 | 0.00 | 0.09 | 0.02 |
| Andradite | 94.77 | 79.94 | 92.77 | 89.60 | 79.84 | 67.14 | 89.59 | 87.69 | 89.44 | 85.92 |
| Pyrope | 0.17 | 0.10 | 0.30 | 0.21 | 0.09 | 0.04 | 0.07 | 0.06 | 0.08 | 0.16 |
| Spessartine | 1.65 | 1.86 | 1.55 | 1.90 | 1.69 | 2.08 | 2.32 | 1.98 | 2.13 | 2.62 |
| Grossular | 2.22 | 18.07 | 5.26 | 7.56 | 18.23 | 30.68 | 6.13 | 10.25 | 7.58 | 9.79 |
| Almandine | 1.03 | 0.00 | 0.12 | 0.73 | 0.04 | 0.00 | 1.82 | 0.02 | 0.68 | 1.49 |

Note: n.d. = not detected; FeO* = total iron content.

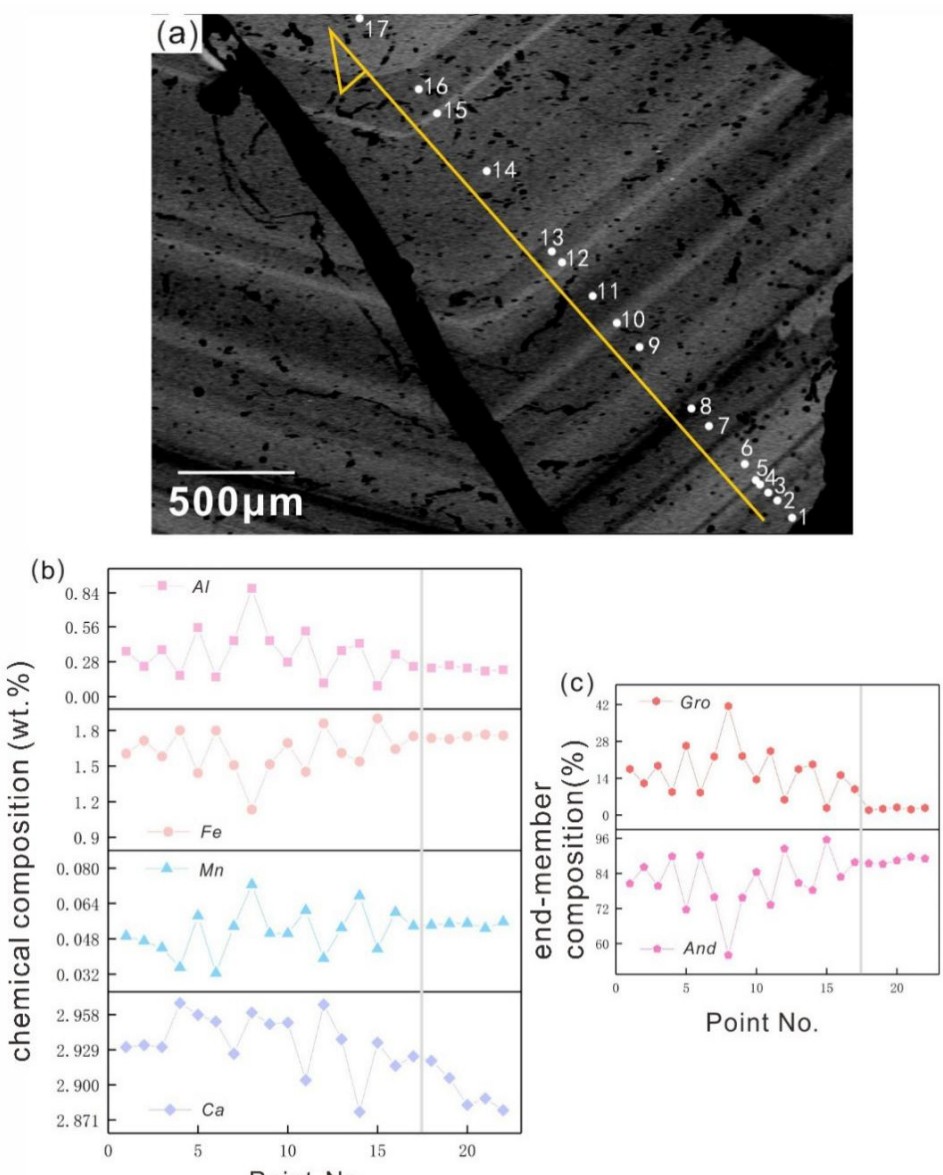

**Figure 5.** (**a**) Backscattered electron image displaying oscillatory zoning in garnet with alternating light and dark domains and the analytical points; (**b**) The major elements of 22 points of the oscillatory zoned rim and core; (**c**) The end-member compositions of 22 points of the oscillatory zoned rim and core. The left side of the vertical gray line is the oscillatory zoned rim, and the right side is the core.

*4.5. Trace Element Geochemistry*

In general, Nanminghe garnets lack large ion lithophile (LIL) elements. Sr, Ba, Cs, and other elements are lower than in the mantle reference, and the LIL elements in some test sites is below the detection limit. The characteristics of rare earth elements in different stages of Nanminghe garnet are slightly different. The garnet in stage I has very obvious positive Eu anomaly and weakly positive Ce anomaly; the stage II garnet has weak Eu anomaly, either positive or negative, and weakly positive Ce anomaly; and the stage III garnet has strong positive Eu anomaly and weak positive Ce anomaly (Table 2). The LREE differentiation degree of early stage garnet is greater than that of late stage garnet. The Eu and Ce anomalies of stage I garnet are more pronounced than those of stage II and III (Figure 6).

**Table 2.** LA–ICP–MS analysis of garnets from the Nanminghe iron deposit (ppm).

| Sample | NSK-J12-1 | NSK-J12-2 | NSK-J12-3 | NSK-J12-4 | NSK-J12-5 | NSK-J12-6 | NSK-J12-7 | NSK-J12-8 | NSK-J12-9 | NSK-J12-10 | NSK-J12-11 | NSK-J12-12 | NSK-J12-13 | NSK-J12-14 |
|---|---|---|---|---|---|---|---|---|---|---|---|---|---|---|
| Stage | | | | | | | II | | | | | | | |
| Sc | 0.44 | 0.64 | 0.7 | 0.83 | 0.47 | 0.79 | 0.64 | 1.1 | 1.05 | 1.01 | 0.79 | 1.16 | 1.26 | 1.16 |
| Sr | 0.33 | 0.28 | 0.28 | 0.26 | 0.8 | 0.33 | 0.6 | 0.37 | 0.33 | 0.34 | 0.47 | 0.33 | 0.34 | 0.43 |
| Y | 3.03 | 67.14 | 2.13 | 66.76 | 3.22 | 78.86 | 18.44 | 17.33 | 10.54 | 6.42 | 7.63 | 10.08 | 9.4 | 9.12 |
| Zr | 0.12 | 1.64 | 1.87 | 5.17 | 0.63 | 4.92 | 1.75 | 3.05 | 3.67 | 2.83 | 2.6 | 3.23 | 3.62 | 3.37 |
| Nb | 0.22 | 9.6 | 2.97 | 25.23 | 5.55 | 20.15 | 5.98 | 5.32 | 2.01 | 1.02 | 0.69 | 0.82 | 0.68 | 0.71 |
| Ba | 0.09 | 0 | 0 | 0 | 0.26 | 0 | 0.49 | 0 | 0 | 0 | 0.3 | 0 | 0 | 0.14 |
| Hf | 0 | 0.01 | 0.04 | 0.08 | 0.02 | 0.06 | 0.05 | 0.07 | 0.14 | 0.13 | 0.08 | 0.12 | 0.15 | 0.13 |
| Ta | 0 | 0.01 | 0.01 | 0.04 | 0 | 0.02 | BDL | 0.01 | 0.01 | 0.03 | 0.02 | 0.03 | 0.03 | 0.03 |
| Pb | 0.19 | 0.12 | 0.04 | 0.05 | 0.16 | 0.13 | 0.26 | 0.06 | 0.04 | 0.09 | 0.18 | 0.07 | 0.06 | 0.09 |
| Th | 7.79 | 3.69 | 2.85 | 3.27 | 2.91 | 8.11 | 7 | 4.25 | 3.06 | 6.05 | 4.09 | 4.27 | 3.99 | 4.34 |
| U | 1.68 | 1.16 | 1.24 | 1.26 | 1.47 | 2.85 | 3.24 | 3.16 | 3.32 | 3.81 | 3.84 | 4.07 | 4.16 | 3.69 |
| La | 2.81 | 1.68 | 1.87 | 1.75 | 4.66 | 3.26 | 3.95 | 4.06 | 3.87 | 4.38 | 4.75 | 4.44 | 4.84 | 4.5 |
| Ce | 22.71 | 16.53 | 18.38 | 18.45 | 22.2 | 29.69 | 34.6 | 34.03 | 32.48 | 38 | 39.29 | 38.72 | 40.47 | 37.36 |
| Pr | 5.68 | 4.79 | 5.04 | 5.72 | 5.91 | 7.24 | 8.16 | 7.94 | 7.65 | 8.17 | 8.25 | 8.36 | 8.54 | 8.15 |
| Nd | 45.46 | 45.53 | 39.24 | 51.82 | 48.92 | 50.44 | 51.11 | 51.43 | 44.05 | 49.12 | 46.28 | 47.16 | 49.73 | 46.88 |
| Sm | 16.78 | 24.57 | 11.93 | 29.23 | 18.1 | 18.3 | 13.87 | 13.59 | 11.95 | 11.79 | 10.17 | 10.65 | 11.52 | 10.37 |
| Eu | 5.76 | 6.47 | 5.2 | 7.97 | 6.5 | 6.24 | 5.82 | 5.9 | 5.12 | 5.19 | 4.88 | 5.24 | 5.14 | 4.85 |
| Gd | 8.21 | 26.3 | 4.77 | 31.69 | 9.74 | 19.45 | 9.26 | 8.54 | 6.8 | 5.79 | 6.4 | 6.86 | 7.03 | 6.71 |
| Tb | 0.55 | 3.32 | 0.27 | 4 | 0.68 | 2.96 | 1.03 | 0.95 | 0.65 | 0.58 | 0.6 | 0.62 | 0.71 | 0.66 |
| Dy | 1.66 | 17.06 | 0.67 | 17.54 | 1.64 | 16.07 | 4.53 | 4.13 | 2.65 | 1.94 | 2.25 | 3 | 2.94 | 3.08 |
| Ho | 0.12 | 2.58 | 0.06 | 2.54 | 0.13 | 3.08 | 0.69 | 0.61 | 0.39 | 0.27 | 0.32 | 0.41 | 0.34 | 0.38 |
| Er | 0.18 | 5.42 | 0.18 | 5.17 | 0.25 | 7.44 | 1.48 | 1.41 | 0.87 | 0.45 | 0.52 | 0.79 | 0.69 | 0.77 |
| Tm | 0.01 | 0.61 | 0.02 | 0.55 | 0.01 | 0.88 | 0.17 | 0.16 | 0.11 | 0.04 | 0.08 | 0.08 | 0.09 | 0.07 |
| Yb | 0.03 | 3.32 | 0.12 | 2.89 | 0.07 | 5.44 | 1.05 | 0.98 | 0.48 | 0.33 | 0.36 | 0.42 | 0.48 | 0.49 |
| Lu | 0.01 | 0.41 | 0.03 | 0.37 | 0.01 | 0.81 | 0.14 | 0.12 | 0.07 | 0.03 | 0.05 | 0.08 | 0.04 | 0.07 |
| ΣREE | 109.96 | 158.58 | 87.79 | 179.68 | 118.82 | 171.31 | 135.87 | 133.83 | 117.17 | 126.1 | 124.2 | 126.82 | 132.57 | 124.33 |
| LREE | 99.19 | 99.56 | 81.66 | 114.94 | 106.29 | 115.17 | 117.51 | 116.94 | 105.13 | 116.67 | 113.62 | 114.56 | 120.25 | 112.11 |
| HREE | 10.77 | 59.02 | 6.13 | 64.75 | 12.54 | 56.14 | 18.36 | 16.89 | 12.03 | 9.43 | 10.58 | 12.26 | 12.32 | 12.22 |
| LREE/HREE | 9.21 | 1.69 | 13.32 | 1.78 | 8.48 | 2.05 | 6.4 | 6.92 | 8.74 | 12.38 | 10.74 | 9.34 | 9.76 | 9.18 |
| $La_N/Yb_N$ | 61.93 | 0.36 | 10.85 | 0.43 | 46.89 | 0.43 | 2.7 | 2.97 | 5.76 | 9.6 | 9.55 | 7.55 | 7.28 | 6.66 |
| δEu | 1.33 | 0.77 | 1.77 | 0.8 | 1.35 | 1 | 1.48 | 1.56 | 1.59 | 1.7 | 1.72 | 1.75 | 1.62 | 1.66 |
| δCe | 1.04 | 0.94 | 0.98 | 0.89 | 0.89 | 1.08 | 1.1 | 1.1 | 1.1 | 1.19 | 1.2 | 1.19 | 1.2 | 1.16 |

| Sample | NSK-22-5 | NSK-22-6 | NSK-22-7 | NSK-22-8 | NSK-22-9 | NSK-22-10 | NSK-22-11 | NSK-22-12 | NSK-22-13 | NSK-22-14 | NSK-22-15 | NSK-22-16 |
|---|---|---|---|---|---|---|---|---|---|---|---|---|
| Stage | | | | | | II | | | | | | |
| Sc | 1.02 | 0.34 | 0.53 | 0.6 | 0.43 | 0.94 | 5.57 | 4.01 | 1.14 | 0.7 | 0.68 | 0.77 |
| Sr | 0.27 | 0.23 | 0.31 | 0.24 | 0.26 | 0.31 | 0.32 | 0.41 | 0.37 | 0.41 | 0.4 | 0.29 |
| Y | 3.67 | 1.2 | 1.67 | 5.54 | 1.57 | 8.72 | 4.14 | 4.73 | 6.77 | 11.09 | 15.83 | 2.67 |
| Zr | 8.45 | 0.11 | 0.9 | 4.9 | 0.34 | 3.13 | 18.86 | 12.66 | 4.15 | 2.37 | 2.45 | 2.09 |
| Nb | 2.02 | 0.26 | 0.84 | 0.08 | 3.49 | 2.01 | 1.32 | 2.73 | 1.33 | 2.74 | 5.01 | 0.51 |

**Table 2.** *Cont.*

| Sample | NSK-22-5 | NSK-22-6 | NSK-22-7 | NSK-22-8 | NSK-22-9 | NSK-22-10 | NSK-22-11 | NSK-22-12 | NSK-22-13 | NSK-22-14 | NSK-22-15 | NSK-22-16 | | |
|---|---|---|---|---|---|---|---|---|---|---|---|---|---|---|
| Stage | | | | | | | II | | | | | | | |
| Ba | BDL | 0.02 | 0.04 | 0.1 | 0.02 | 0 | 0 | 0.08 | 0.02 | 0.02 | 0.04 | 0 | | |
| Hf | 0.12 | 0.01 | 0.01 | 0.1 | 0.02 | 0.04 | 0.73 | 0.58 | 0.09 | 0.06 | 0.04 | 0.06 | | |
| Ta | 0 | 0 | 0 | 0.01 | 0 | 0.04 | 0.1 | 0.11 | 0.06 | 0.02 | 0.02 | 0.02 | | |
| Pb | 0.08 | 0.07 | 0.06 | 0.02 | 0.05 | 0.05 | 0.07 | 0.1 | 0.08 | 0.03 | 0.07 | 0.1 | | |
| Th | 1.58 | 2.85 | 3.22 | 1.13 | 3.64 | 2.76 | 2.47 | 3.13 | 5.2 | 2.3 | 4.04 | 5.31 | | |
| U | 1.19 | 1.24 | 1.66 | 3.18 | 1.6 | 2.55 | 3.79 | 3.27 | 3.06 | 2.65 | 2.8 | 2.64 | | |
| La | 2.81 | 2.45 | 2.69 | 0.85 | 2.22 | 3.25 | 4.83 | 3.98 | 3.61 | 3.38 | 3.61 | 3.1 | | |
| Ce | 18.95 | 18.13 | 22.36 | 19.21 | 22.02 | 29.54 | 38.64 | 34.59 | 31.87 | 29.8 | 32.02 | 31.3 | | |
| Pr | 3.48 | 4.54 | 5.42 | 7.27 | 6.1 | 7.01 | 7.78 | 7.31 | 7.8 | 7.18 | 7.21 | 7.22 | | |
| Nd | 24.2 | 32.84 | 38.78 | 47.6 | 51.05 | 46.77 | 40.07 | 39.97 | 48.66 | 45.46 | 41.71 | 47.01 | | |
| Sm | 7.87 | 8.66 | 7.41 | 4.94 | 17.26 | 12.81 | 7.09 | 7.91 | 12.83 | 11.99 | 11.33 | 10.67 | | |
| Eu | 2.74 | 4.12 | 4.08 | 4.76 | 6.76 | 5.64 | 4.11 | 4.16 | 5.62 | 5.37 | 5.04 | 5.32 | | |
| Gd | 4.31 | 2.98 | 2.19 | 1.38 | 7.68 | 7.34 | 3.56 | 3.89 | 6.51 | 7.88 | 8.34 | 4.31 | | |
| Tb | 0.34 | 0.17 | 0.14 | 0.19 | 0.45 | 0.73 | 0.29 | 0.33 | 0.62 | 0.86 | 0.83 | 0.37 | | |
| Dy | 1.16 | 0.4 | 0.48 | 0.95 | 0.89 | 2.86 | 1.06 | 1.4 | 2.2 | 3.19 | 4.3 | 1.01 | | |
| Ho | 0.12 | 0.05 | 0.06 | 0.22 | 0.06 | 0.35 | 0.14 | 0.18 | 0.27 | 0.44 | 0.6 | 0.09 | | |
| Er | 0.35 | 0.1 | 0.18 | 0.63 | 0.07 | 0.62 | 0.29 | 0.38 | 0.44 | 0.76 | 1.29 | 0.14 | | |
| Tm | 0.03 | 0.01 | 0.02 | 0.19 | 0.01 | 0.04 | 0.05 | 0.05 | 0.04 | 0.08 | 0.15 | 0.01 | | |
| Yb | 0.33 | 0.08 | 0.11 | 1.66 | 0.03 | 0.33 | 0.22 | 0.31 | 0.2 | 0.48 | 0.88 | 0.09 | | |
| Lu | 0.05 | 0.01 | 0.03 | 0.29 | 0 | 0.04 | 0.05 | 0.05 | 0.02 | 0.05 | 0.11 | 0.02 | | |
| ΣREE | 66.74 | 74.53 | 83.95 | 90.16 | 114.62 | 117.33 | 108.18 | 104.49 | 120.68 | 116.92 | 117.42 | 110.65 | | |
| LREE | 60.05 | 70.74 | 80.74 | 84.63 | 105.42 | 105.02 | 102.52 | 97.92 | 110.39 | 103.18 | 100.93 | 104.61 | | |
| HREE | 6.69 | 3.79 | 3.21 | 5.53 | 9.2 | 12.31 | 5.66 | 6.57 | 10.29 | 13.74 | 16.49 | 6.04 | | |
| LREE/HREE | 8.98 | 18.66 | 25.12 | 15.31 | 11.46 | 8.53 | 18.12 | 14.9 | 10.72 | 7.51 | 6.12 | 17.31 | | |
| $La_N/Yb_N$ | 6.06 | 23.47 | 17.81 | 0.37 | 52.57 | 7.02 | 16.06 | 9.27 | 13.04 | 5.02 | 2.96 | 24.64 | | |
| δEu | 1.31 | 2 | 2.38 | 4.21 | 1.55 | 1.63 | 2.23 | 2.03 | 1.68 | 1.59 | 1.52 | 2.02 | | |
| δCe | 1.28 | 1.02 | 1.07 | 0.78 | 0.98 | 1.1 | 1.23 | 1.21 | 1.07 | 1.08 | 1.15 | 1.15 | | |

| Sample | NSK-J5-1 | NSK-J5-2 | NSK-J5-3 | NSK-J5-4 | NSK-J5-5 | NSK-J5-6 | NSK-J5-2-1 | NSK-J5-2-2 | NSK-J5-2-3 | NSK-J5-2-4 | NSK-J22-1 | NSK-J22-2 | NSK-J22-3 | NSK-J22-4 |
|---|---|---|---|---|---|---|---|---|---|---|---|---|---|---|
| Stage | | | | | | I | | | | | | III | | |
| Sc | 0.34 | 0.51 | 0.38 | 0.5 | 0.35 | 0.43 | 0.36 | 0.41 | 0.32 | 0.43 | 0.95 | 1.05 | 0.73 | 0.73 |
| Sr | 0.94 | 0.47 | 0.54 | 0.61 | 0.59 | 0.6 | 0.61 | 0.6 | 0.72 | 0.5 | 0.17 | 0.23 | 0.2 | 0.15 |
| Y | 0 | 1.63 | 0.03 | 0.05 | 0 | 0.01 | 0.01 | 0.01 | 0.02 | 0.01 | 15.06 | 97.6 | 16.75 | 154.22 |
| Zr | 0 | 2.55 | 0.07 | 0.08 | BDL | 0.05 | 0.08 | 0.03 | 0.05 | 0.08 | 17.85 | 19.75 | 14.69 | 19.33 |
| Nb | 0 | 0.48 | 0.01 | 0 | 0 | 0 | 0.01 | 0 | 0.01 | 0 | 1 | 5.76 | 0.52 | 10.61 |
| Ba | BDL | 0.26 | 0 | BDL | 0.04 | 0 | 0 | 0 | 0.04 | 0 | BDL | 0.19 | 0.06 | 0.02 |
| Hf | 0 | 0.04 | 0 | 0.01 | 0 | 0 | 0 | 0 | 0 | 0.01 | 0.43 | 0.7 | 0.35 | 0.66 |
| Ta | 0 | 0.03 | 0 | 0 | 0 | 0 | 0 | 0 | 0.01 | 0 | 0.03 | 0.07 | 0.01 | 0.16 |
| Pb | 0.06 | 0.14 | BDL | 0.02 | 0.02 | BDL | 0.02 | BDL | 0.05 | 0.02 | 0.05 | 0.01 | BDL | 0.01 |
| Th | 0.01 | 2.83 | 0.13 | 0.21 | 0.1 | 0.13 | 0.11 | 0.42 | 3.15 | 0.42 | 2.48 | 1.18 | 1.66 | 0.72 |

**Table 2.** *Cont.*

| Sample | NSK-J5-1 | NSK-J5-2 | NSK-J5-3 | NSK-J5-4 | NSK-J5-5 | NSK-J5-6 | NSK-J5-2-1 | NSK-J5-2-2 | NSK-J5-2-3 | NSK-J5-2-4 | NSK-J22-1 | NSK-J22-2 | NSK-J22-3 | NSK-J22-4 |
|---|---|---|---|---|---|---|---|---|---|---|---|---|---|---|
| Stage | I | | | | | | | | | | III | | | |
| U | 21.82 | 4.5 | 4.58 | 9.63 | 12.14 | 13.28 | 13.73 | 10.8 | 29.53 | 11.31 | 3.33 | 1.98 | 1.94 | 1.5 |
| La | 91.79 | 8.79 | 6.52 | 14.11 | 15.66 | 18.45 | 20.97 | 16.96 | 42.48 | 14.49 | 0.89 | 0.39 | 0.52 | 0.23 |
| Ce | 120.89 | 54.68 | 46.81 | 84.03 | 90.92 | 102.83 | 104.53 | 95.77 | 192.87 | 86.31 | 12.9 | 6.78 | 9.32 | 4.52 |
| Pr | 5.3 | 12.33 | 6.73 | 10.94 | 12.28 | 13 | 12.38 | 13.31 | 22.93 | 12.32 | 4.73 | 2.72 | 3.51 | 1.9 |
| Nd | 4.27 | 49.12 | 19.12 | 27.85 | 29.11 | 30.36 | 27.96 | 43.22 | 63.44 | 37.56 | 40.45 | 24.05 | 31.99 | 19.88 |
| Sm | 0.03 | 1.14 | 0.26 | 0.41 | 0.34 | 0.33 | 0.24 | 1.29 | 1.66 | 0.91 | 11.61 | 12.98 | 10.24 | 13.92 |
| Eu | 0.29 | 2.06 | 1.6 | 2.01 | 2.32 | 2.26 | 1.93 | 3.42 | 3.85 | 3.04 | 16.83 | 17.49 | 16.86 | 18.31 |
| Gd | 0.05 | 0.41 | 0.04 | 0.04 | 0.08 | 0.08 | 0.1 | 0.14 | 0.14 | 0.08 | 5.65 | 17.83 | 6.17 | 23.95 |
| Tb | 0 | 0.05 | 0 | 0 | 0 | 0 | 0 | 0 | BDL | 0 | 0.51 | 2.91 | 0.55 | 4.52 |
| Dy | 0 | 0.29 | 0 | 0.01 | BDL | 0 | 0 | 0 | 0.01 | 0 | 2.66 | 18.13 | 2.97 | 28.34 |
| Ho | 0 | 0.05 | 0 | 0 | 0 | 0 | 0 | 0 | 0 | 0 | 0.56 | 3.79 | 0.53 | 6.1 |
| Er | 0 | 0.15 | 0 | BDL | 0 | 0 | 0 | 0 | 0 | 0 | 2.07 | 10.65 | 2.29 | 15.87 |
| Tm | 0 | 0.02 | 0 | 0 | 0 | 0 | 0 | 0 | 0 | 0 | 0.59 | 1.79 | 0.71 | 2.48 |
| Yb | 0 | 0.07 | 0 | 0 | 0.01 | 0 | 0 | 0 | 0.01 | 0 | 9.21 | 16.38 | 9.75 | 21.92 |
| Lu | 0 | 0.02 | 0 | 0 | BDL | BDL | 0 | 0 | 0 | 0 | 2.13 | 3.54 | 2.41 | 4.69 |
| ΣREE | 222.64 | 129.17 | 81.08 | 139.4 | 150.72 | 167.31 | 168.1 | 174.13 | 327.39 | 154.72 | 110.79 | 139.44 | 97.81 | 166.62 |
| LREE | 222.58 | 128.11 | 81.04 | 139.35 | 150.63 | 167.23 | 168 | 173.97 | 327.23 | 154.63 | 87.4 | 64.41 | 72.44 | 58.75 |
| HREE | 0.06 | 1.06 | 0.04 | 0.05 | 0.09 | 0.08 | 0.1 | 0.16 | 0.16 | 0.08 | 23.39 | 75.03 | 25.38 | 107.88 |
| LREE/HREE | 3790.46 | 121.09 | 1835.67 | 2810.74 | 1678.55 | 2026.07 | 1690.3 | 1070.34 | 2060.97 | 1819.49 | 3.74 | 0.86 | 2.85 | 0.54 |
| $La_N/Yb_N$ | | 90.15 | | | 1677.7 | | | | 1780.61 | 4501.45 | 0.07 | 0.02 | 0.04 | 0.01 |
| δEu | 20.53 | 7.5 | 29.56 | 23.87 | 30.77 | 30.82 | 32.48 | 12.92 | 11.5 | 16.57 | 5.61 | 3.51 | 6 | 3.04 |
| δCe | 0.89 | 1.07 | 1.56 | 1.57 | 1.52 | 1.57 | 1.56 | 1.48 | 1.5 | 1.48 | 0.79 | 0.73 | 0.78 | 0.7 |

Note: BDL = Below detection limit.

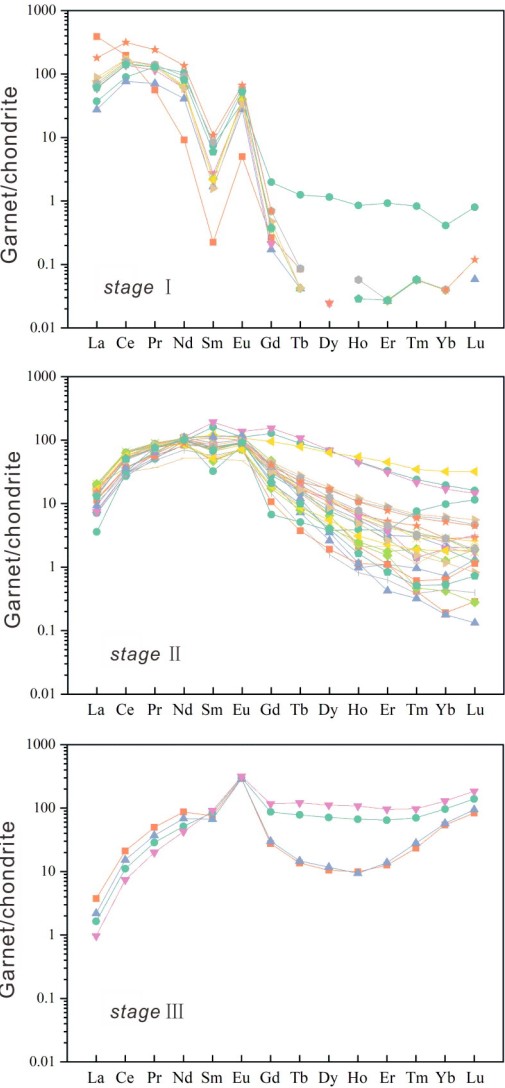

**Figure 6.** Chondrite-normalized REE patterns of the garnets from the Nanminghe iron deposit.

## 5. Discussion

### 5.1. Origin of Oscillatory Zoning

Garnets in skarn deposits commonly display oscillatory zoning patterns such as chemical zoning, reflecting the changes in physicochemical conditions during garnet crystallization and/or changing fluid compositions and properties during subsequent growth [7,29–31]. As Ranjbar et al. (2016) [32] reported, oscillatory zoning may indicate oscillatory changes in fluid composition under internal and/or external control. Tian et al. (2019) [12] divided the various theories on the origin of oscillatory zoning into two different categories: (1) Internal crystal growth processes, such as the interplay of chemical diffusion, thermal diffusion, and rates of crystal growth [33,34], and (2) cyclic variations in external factors [11,35,36], such as multiple pulses of magmatic fluid [37] or repeating crack-seal processes of the hydrothermal system [12].

Oscillatory zoned garnets from the Nanminghe iron deposit are characterized by andradite-rich cores and grossular–andradite rims, which were similar to the garnets from the Kasva skarn [29] but are different from many skarn garnets with grossular-rich cores and grossular–andradite rims related to Fe deposits worldwide [5,20,30,38]. Mirnejad et al. (2018) [29] proposed that hot magmatic fluid with high $fO_2$ can lead to the core-to-rim growth variations. Pertoldova et al. (2009) [9] suggested that the presence of magnetite and the increasing proportion of the andradite component in the garnet indicate that the iron is mostly in the form of $Fe^{3+}$ in high $fO_2$ condition, which favors andradite

precipitation [12,20,36,39]. Jamtveit et al. (1993) [25] proposed that boiling, commonly associated with periods of fluid flow in hydrothermal systems, caused the higher $fO_2$ and $Fe^{3+}$ and corresponding andradite content. When boiling ceased, the interaction between fluid and wall rock tends to reduce the fluid back to an $fO_2$ value that is locally buffered by $Fe^{2+}$ bearing silicates and favors grossular precipitation [40]. The breccia commonly occurring in the Nanminghe iron deposit and the numerous cracks under the microscope may be the indication of hydrofracturing after boiling [41].

Tian et al. (2019) [12] discussed the formation of oscillatory zoning in garnets from the Hongshan Cu-Mo skarn deposit and proposed that the Al-rich zones were likely formed under a relatively reduced and near-neutral pH condition, whereas the Fe-rich zones probably formed in a more oxidized and mildly acidic condition. Accordingly, for the Nanminghe skarn garnets, it is suggested that the hydrothermal fluids from which the andradite-rich cores formed might have been mildly acidic and reduced, whereas those for the oscillatory-zoned grossular–andradite rims experienced periodic variations between relatively neutral and oxidized fluids and mildly acidic and reduced fluids [8,29].

*5.2. Formation of Andradite-Rich Garnet*

Demir et al. (2017) [41] proposed that the andradite-rich garnet and the presence of magnetite-dominated ore can indicate oxidized-type mineralization [42]. Most of the Fe ions are ferric, indicating that the formation of the andradite occurred under oxidizing conditions [43]. It is suggested that in a high $fO_2$ condition, the stage I garnets and the core of stage II garnets are precipitated in the early stage of garnet formation. According to previous studies, the formation of garnet leads to the increase in porosity, the release of pressure in the system, triggers fluid boiling, and results in the periodic change in fluid composition and the external environment. The rim of stage II euhedral garnet developed oscillatory zoning. Late stage III veined garnets are formed in the retrograde stage of skarn formation from the previous existence of garnet by later fluid recrystallization along the fissure, cutting stage II garnets. The proportions of andradite of garnet in stage I to III show a gradual decrease, which can be due to the iron ions in the hydrothermal fluid accumulating and precipitating to form magnetite with the decrease in temperature, rather than forming andradite-rich domains by metasomatism with surrounding rocks. The dispersed magnetite filling intergranular space and cracks in stage I garnets indirectly indicates that the formation of garnet provided the conditions for magnetite precipitation [44]. The hydrothermal origin of Nanminghe skarn garnets is also supported by oscillatory zoning, minerals' inclusions such as magnetite and diopside, and fluid inclusions [45].

The mechanism of trace elements entering crystals is mainly surface adsorption (mainly controlled by electrostatic interaction between minerals) and isomorphic substitution (mainly controlled by crystal chemistry) [24]. Due to the small radius of HREE, it is easier to enter the crystal structure of garnet by replacing the calcium ions with eight-coordination. Most of the metamorphic and magmatic aluminum series garnets show a left-inclined distribution pattern of heavy rare earth enrichment and light rare earth depletion because their REE distribution is mainly controlled by crystal chemistry. However, the garnet of skarn has a variety of rare earth distribution patterns, which indicates that the rare earth distribution pattern of calcium garnet in skarn is not only controlled by crystal chemistry but also affected by the electrostatic interaction between minerals. Therefore, it is speculated that the rare earth elements in Nanminghe garnet mainly enter the garnet crystal through adsorption.

The Nanminghe iron deposit is closely related to Yanshanian magmatic rocks in genesis. Ren et al. (2010) [46] mentioned that the distribution pattern of rare earth in a magmatic hydrothermal system is usually right-leaning, accompanied by Eu anomaly. It is speculated that REE in Nanminghe garnet may come from magmatic fluid and enter into the garnet's structure mainly through adsorption mechanism. The overall rare earth distribution pattern of Nanminghe garnet is consistent with that of andradite in the Crown Jewel gold deposit in North America [24]. Gaspar et al. (2008) [24] argue that this kind of

REE distribution pattern occurs when andradite is formed by metasomatism at relatively high water rock ratios and that $Cl^-$ plays an important role in the transport of REE; this type of garnet often shows oscillation bands and has faster growth rates, with visible tetragonal trioctahedral crystal planes. This is consistent with the mineralogical characteristics of Nanminghe garnet. According to previous studies, $F^-$, $Cl^-$, and $OH^-$ are all possible carriers of REE. The electron probe test data of some Nanminghe garnet showed the presence of $Cl^-$, but did not detect $F^-$, and the infrared spectrum did not detect the presence of $OH^-$, so the chlorine complexes may play a role in the formation of Nanminghe garnet.

## 6. Conclusions

1. The brown to dark brown garnet in the Nanminghe skarn iron deposit is characterized by a combination of dodecahedral and trapezohedron faces, and the associated minerals include diopside, magnetite, quartz, calcite, and pyrite.

2. Based on petrographic and compositional characteristics, we identify at least three stages of garnet growth. From early to late stage garnets, the iron content gradually decreases, which provides the necessary ingredients and growth space for the precipitation of magnetite. The iron ore is a product of contact metasomatism, precipitated in the contact zone between Middle Ordovician limestones and Yanshanian diorite.

3. Our results, in conjunction with those from previous studies, indicate that the rare earth elements may mainly come from magmatic hydrothermal fluids, and chloride complexes play a certain role in the material transportation. Nanminghe garnet may be formed under infiltration metasomatism, with a high water–rock ratio and a high oxygen fugacity in the formation environment. It is speculated that garnet is the product of hydrothermal fluid and provides conditions for the later magnetite precipitation.

**Author Contributions:** Validation, C.-T.R. and X.-Y.Y.; formal analysis, C.-T.R. and X.-Y.Y.; investigation, C.-T.R. and X.-Y.Y.; resources, S-G.S.; data curation, C.-T.R.; writing—original draft preparation, C.-T.R.; writing—review and editing, C.-T.R., X.-Y.Y., S-G.S., L.-J.Q. and M.S.; funding acquisition, X.-Y.Y. All authors have read and agreed to the published version of the manuscript.

**Funding:** This research was funded by the China Geological Survey Project "Geology of Mineral Resources in China" (Grant No: DD20190379-88) to Prof. Xiao-Yan Yu and the Major Research Plan of the Natural Science Foundation of China (Grant No:92162213) to Prof. Shang-Guo Su.

**Data Availability Statement:** All data generated or used during the study appear in the submitted article.

**Acknowledgments:** We thank Jizhong Energy Fengfeng Group Wu'an Nanminghe Iron Mine Co., Ltd. for providing samples. We gratefully acknowledge Xin Lu and Xiao-Yang Hou for their help in sample collection and data analysis. We appreciate Ming Zhang and Zhu-Lin Sun for their assistance in the IR and Raman experiments, respectively.

**Conflicts of Interest:** The authors declare no conflict of interest.

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
