# Peer review of "Anatomy of Garnet from the Nanminghe Skarn Iron Deposit, China: Implications for Ore Genesis"

_minerals, doi:10.3390/min12070845_

Round 1
Reviewer 1 Report
Dear Authors,
This paper is a good quality and well writing. But, some references (especially Ranjbar et al. 2016 for hydrothermal system) need to add and discuss to the discussion section (origin of oscillatory zoning and formation of andradite-rich garnet) in the article. Garnet composition samples from literature (Ranjbar et al. 20016 for hydrothermal system; Sipahi et al. 2017 for skarn system; Demir et al. 2017 for skarn system; Sipahi et al. 2021 for skarn system; Duan et al. 2020 etc.) should be added to Figure 6. These references will enrich the content of article.
Reviewer 2 Report
RE: Anatomy of garnet from the Nanminghe skarn iron deposit, China: Implications
for the genesis of iron mineralization
In this paper, the authors provide very good and complete information about the distribution trend of antimony in the garnet mineral from Nanminghe skarn iron deposit, China and use it as an effective geochemical parameter in determining the genesis of iron ore. This article seems to be very well organized and has high quality. Therefore, this article is acceptable in the journal with minor revisions:
Comment 1:
It is better to change the title of the article as follows:
Anatomy of garnet from the Nanminghe skarn iron deposit, China: Implications
for ore genesis
Comment 2:
Please show the names of the minerals on the peaks in Figure 5. In its current form, the reader has no understanding of these XRD graphs.
Reviewer 3 Report
Summary
The paper provides various spectroscopic and chemical data on garnet from the Nanminghe Fe skarn deposit, China. The authors interpret different generations of garnet formation to represent stages of growth and mineralisation in the deposit.
General comments
The introduction and background sections are sufficient and well written.
The authors use the word “representative” throughout when referring to their samples, however with only 7 samples analysed via microscopic methods, it is hard to justify this. Generally speaking, skarn deposits are extremely variable and heterogeneous, so you would need much more than 24 or 7 samples for a complete study.
Moreover, the authors mention there are six different skarn ore bodies at the deposit, but there is no information on which ore bodies their samples are from. There is also no sample list/table that shows the sample depth and location. Given that the deposit varies in depth from 80-800m, this has further implications for the discussion section, particularly the evolution of the deposit. If the authors have only sampled a few parts of the deposit then surely the chemistry of the garnet would differ between the shallow and deeper ore bodies. This might not be obvious in major elements, but certainly trace elements would show great variation due to differences in physicochemical conditions of formation.
The authors do not specify whether the skarn is endo or exo, merely at the contact. This is important because there will be significant geochemical and textural differences between endo and exo skarns. They might also sample the unaltered host rocks to determine metal sources as well as potential mass balance. There is no mention of prograde and retrograde timing of crystallization, which is fundamental in skarn deposits.
More detail needs to be added to the Methodology section. Particularly for the microprobe, such as minimum detection limits etc. The authors should consider performing LA-ICP-MS if possible. This would help constrain trace element differences between their garnet generations and provide some useful insight into fluid chemistry/evolution. There are numerous papers on modelling of trace elements in garnets from skarns.
Results sections 4.3, 4.4 and 4.5 can be shortened and combined into one section. There is no need to state all the chemical compositions, which are listed in the table (e.g. section 4.5).
Small inclusions were observed in the garnets when using Raman, however no detail is given as to the speciation of these phases. If they are not salts, then Raman should be able to identify them. Also how many different types of inclusions have been found? The authors could consider performing a fluid inclusion study on their garnets too. Given the various depths of the skarn bodies, there will certainly be differences in homogenisation temperatures across the deposit.
The use of Infrared spectroscopy here is a bit puzzling. It doesn’t really add anything to the paper. Major element chemistry is enough to calculate end member compositions for garnet. The authors could simply state that based on infrared no OH was found. There is no need for a whole section in the discussion to explain this (5.1). Moreover, in this section, there is very little discussion on the actual chemistry of garnet. It is mainly focussed on spectroscopic findings, which add very little to the paper. I would consider removing this section altogether.
In Discussion 5.2 the authors simply provide a weak summary of what other papers have discussed regarding oscillatory zonation in garnet. Their arguments are speculative and not very well constrained. The lack of representative samples also brings into question the validity of these interpretations.
Discussion 5.3 again is brief. In line 309 “the formation of garnet leads to increase of porosity”, of what? I’m not sure what the authors mean here. Also the interpretation of “vein stage” garnet is most likely just recrystallization of pre-existing garnet from later fluids, perhaps during the retrograde stage of skarn formation.
Reading further in this section it appears that trace elements and fluid inclusions have already been done for garnets from this deposit (again we don’t know from which of the six ore bodies though). This begs the question as to what new findings the authors are actually presenting in this paper? We know that different garnet end members have different spectroscopic characteristics, and the interpretations of different redox conditions is speculative and based purely on previous studies. For these reasons, I have to suggest the authors reconsider their study objectives, obtain more representative samples and, if possible, perform additional measurements to support any genetic interpretations.
Figures
Figures 2 and 3 can be combined with fewer images. Many of them are unnecessary.
Figure 6 is very poorly presented and doesn’t provide much to the paper. Aside from the aesthetics, there is no detail as to which skarn deposits are used in their classification.
The BSE imagine in Figure 7 needs to be enlarged.
Round 2
Reviewer 3 Report
N/A
Author Response
Thank you for your precious review. It has been revised according to the suggestions of academic editor. For details, please refer to the manuscript submitted after modification.